# Study on Antidepressant Activity of Pseudo-Ginsenoside HQ on Depression-Like Behavior in Mice

**DOI:** 10.3390/molecules24050870

**Published:** 2019-03-01

**Authors:** Li-xue Chen, Zeng Qi, Zi-jun Shao, Shan-shan Li, Yu-li Qi, Kun Gao, Song-xin Liu, Zhuo Li, Yin-shi Sun, Ping-ya Li

**Affiliations:** 1Institute of Special Animals and Plants Sciences, Chinese Academy of Agricultural Sciences, Changchun 130112, China; 82101172456@caas.cn (L.-x.C.); shaozijun2017@163.com (Z.-j.S.); Lishanshan@caas.cn (S.-s.L.); 15568871905@163.com (Y.-l.Q.); GaoK2018@163.com (K.G.); liusx2018@163.com (S.-x.L.); 2College of Chinese Medicinal Materials, Jilin Agricultural University, Changchun 130118, China; 3School of Pharmaceutical Sciences, Jilin University, Changchun 130021, China; qizeng95@163.com (Z.Q.); zhuoli198602@gmail.com (Z.L.)

**Keywords:** ginsenoside Rh_2_, (24*R*)-pseudo-ginsenoside HQ, (24*S*)-pseudo-ginsenoside HQ, LPS-induced depression

## Abstract

Suppressive effects of ginsenoside Rh_2_ (Rh_2_), (24*R*)-pseudo-ginsenoside HQ (*R*-PHQ), and (24*S*)-pseudo-ginsenoside HQ (*S*-PHQ) against lipopolysaccharide (LPS)-induced depression-like behavior were evaluated using the forced swimming test (FST) and tail suspension test (TST) in mice. Pretreatment with Rh_2_, *R*-PHQ, and *S*-PHQ significantly decreased immobility time in FST and TST with clear dose-dependence, and significantly downregulated levels of serum tumor necrosis factor-α and interleukin-6, and upregulated superoxide dismutase activity in the hippocampus of LPS-challenged mice. Furthermore, *R*-PHQ and *S*-PHQ significantly increased the expression of the brain-derived neurotrophic factor (BDNF), tropomyosin-related kinase B (TrkB), sirtuin type 1 (Sirt1), and nuclear-related factor 2, and inhibited the phosphorylation of inhibitor of κB-α and nuclear factor-κB (NF-κB) in the hippocampus of LPS-challenged mice. Additionally, the antidepressant-like effect of *R*-PHQ was found related to the dopaminergic (DA), γ-aminobutyric acid (GABA)ergic, and noradrenaline systems, while the antidepressive effect of *S*-PHQ was involved in the DA and GABAergic systems. Taken together, these results suggested that Rh_2_, *R*-PHQ, and *S*-PHQ produced significant antidepressant-like effects, which may be related to the BDNF/TrkB and Sirt1/NF-κB signaling pathways.

## 1. Introduction

Major depressive disorder (MDD) is a common psychiatric disorder, which is widely distributed in the population [1,2]. Although the improvements in treatment and research are encouraging, depression is still a serious problem that could lead to serious health complications, even putting the patients’ lives at risk [3]. More than 17% of people worldwide suffer from depression, and around one million depressed patients commit suicide every year [4].

Exposure to the bacterial endotoxin lipopolysaccharide (LPS) can induce depressive symptoms in rodents and humans, and the LPS-induced immunoreactive mice model is a recognized inflammatory-related depressive animal model [5]. LPS challenge could cause the activation of innate immune response and promote the secretion of proinflammatory cytokines, such as serum tumor necrosis factor-α (TNF-α) and interleukin (IL)-6 [6]. LPS can activate mitogen-activated protein kinases (MAPK) in cells, or nuclear factor signal transduction pathways and corresponding transcription factors. This provided reason to regulate inflammatory mediators via the expression of IL-1, TNF-α, and so on, further releasing nitric oxide, tumor necrosis factor, interleukin, free radicals, and a large number of toxic cytokines, resulting in the body suffering oxidative stress and inflammation reactions [7,8]. These cytokines could further result in oxidative stress, characterized by suppressed superoxide dismutase (SOD) activity and elevated malondialdehyde level in the hippocampus, which are considered as the mechanisms of LPS-induced depression [9].

For most physiological illnesses, antidepressants are the first line of medical treatment of MDD. However, clinical use of commercially available antidepressants is often accompanied by some side effects and adverse reactions, such as weight gain, dry mouth, sexual problems, and nausea, which may add another medical condition and lessen the willingness of patients during drug therapy [3]. Considering this phenomenon, researchers show great interest in discovering, studying, and using more natural medicines for remedy [10,11].

Ginseng, the root of *Panax ginseng* Meyer, was a widely used folk medicine in many Asian countries including China, Korea, and Japan for a thousand years [12,13]. It was reported that ginseng extract G115, the standardized extract of *Panax ginseng*, exhibits an antidepressant effect in ethanol-treated mice models via increasing brain-derived neurotrophic factor (BDNF) levels in the hippocampus and prefrontal cortex [14]. Ginsenosides are the main active ingredient of ginseng [15,16], whereby some ginsenosides such as ginsenoside Rg_3_, Rg_5_, and Rh_3_ showed antidepressant-like effects through the hippocampus BDNF signaling pathway [6,17,18]. It was also reported that ginsenoside Rh_2_ exerted anxiolytic-like effects in the elevated plus-maze model [19]. However, the effect of Rh_2_ on the LPS-induced depressive animal model remains poorly understood. In this study, we investigate the effects of Rh_2_ and its metabolites on LPS-induced depression in mice for the first time.

## 2. Results

### 2.1. Open-Field Test

To avoid false positive results in the behavioral despair test, the effects of treated drugs on locomotor activity in mice were tested (Figure 1A). The results showed that LPS exposure, or drug administration of fluoxetine (FLU), Rh_2_, (24*R*)-pseudo-ginsenoside HQ (*R*-PHQ), or (24*S*)-pseudo-ginsenoside HQ (*S*-PHQ) led to no significant effects on the autonomous activity of mice as compared to the control group.

### 2.2. Effects of Rh_2_, R-PHQ, and S-PHQ on LPS-Induced Depressive-Like Behavior

The forced swimming test (FST) and tail suspension test (TST) are two classical behavioral models to evaluate the antidepressive activity of drugs [20,21]. As compared to the control group, LPS challenge led to an evident increase of immobility time in both TST and FST (Figure 1B,C) (*p <* 0.05, *p <* 0.01). However, pretreatment with FLU, Rh_2_, *R*-PHQ, or *S*-PHQ alleviated the depressive symptoms. Rh_2_, *R*-PHQ, and *S*-PHQ exhibited dose-dependent antidepressive effects in TST. Rh_2_, *R*-PHQ, and *S*-PHQ at a dose of 30 mg/kg showed the highest antidepressive effects in both TST and FST for corresponding drugs, and, among them, *R*-PHQ showed relative better antidepressive effects. Western blot showed LPS exposure induced downregulation of neurotrophic factors BDNF and tropomyosin-related kinase B (TrkB); however, Rh_2_, *R*-PHQ, and *S*-PHQ treatment recovered the expressions of BDNF and TrkB (Figure 2).

### 2.3. Rh_2_, R-PHQ, and S-PHQ Mediate LPS-Induced Inflammatory Reaction

LPS-induced pathophysiology is associated with inflammation. In this study, we also observed obviously higher levels of serum TNF-α and IL-6 in the LPS group compared to the control group. However, pretreatment with FLU, Rh_2_, *R*-PHQ, and *S*-PHQ clearly decreased the serum TNF-α and IL-6 levels compared to the LPS group (Figure 3A,B). A similar phenomenon was observed in mechanism analysis, with a significant upregulation of phosphorylation of inflammation-associated protein inhibitor of κB-α (IκB-α) and nuclear factor-κB (NF-κB) in the hippocampus tissues of the LPS group observed as compared to the control. In contrast, Rh_2_, *R*-PHQ, and *S*-PHQ significantly inhibited the phosphorylation of IκB-α and NF-κB (Figure 2).

### 2.4. Effects of Rh_2_, R-PHQ, and S-PHQ on LPS-Induced Oxidative Stress in the Hippocampus

Previous studies also reported the involvement of the antioxidant system in the LPS-caused depressant-like behaviors [22]. As compared to the control group, LPS exposure obviously decreased the SOD activity, but pretreatment with Rh_2_, *R*-PHQ, and *S*-PHQ relatively recovered the level of SOD in the hippocampus (Figure 3C). In terms of expression anti-oxidant-associated proteins, consistent with the effect on the SOD activity, Rh_2_, *R*-PHQ, and *S*-PHQ recovered the LPS-induced decrease of sirtuin type 1 (Sirt1) protein expression, but Rh_2_ exhibited a moderate effect on nuclear-related factor 2 (Nrf2), while *R*-PHQ and *S*-PHQ clearly elevated the protein expression of Nrf2 in the hippocampus (Figure 2).

### 2.5. Role of the Different Facets of the Central Nervous System including Dopaminergic (DA), γ-Aminobutyric Acid (GABA)ergic, and Noradrenergic (NA) Systems in the Antidepressant-Like Effect of R-PHQ and S-PHQ in the FST

In the concurrent studies, three agonists haloperidol, bicuculline, and prazosin were combined with administration of *R*-PHQ or *S*-PHQ in the FST to verify if any central nervous system (CNS) facets participate in the antidepressant-like effect of *R*-PHQ and *S*-PHQ. To confirm the action pathway of *R*-PHQ and *S*-PHQ, we measured the levels of DA, GABA, and NA systems in mice brain.

#### 2.5.1. Involvement of the Dopaminergic System in the Antidepressant-Like Effect of *R*-PHQ and *S*-PHQ in the FST

As compared to the control group, both *R*-PHQ and *S*-PHQ treatment groups exhibited shorter immobility time in the FST (Figure 4A). However, the *R*-PHQ- and *S*-PHQ-induced reduction of immobility time in the FST was partly eliminated by pre-treatment with haloperidol. Both *R*-PHQ and *S*-PHQ could increase the content of DA in mice brain, while the dopaminergic system antagonist haloperidol attenuated these increasing trends (Figure 4B).

#### 2.5.2. Involvement of the GABAergic System in the Antidepressant-Like Effect of *R*-PHQ and *S*-PHQ in the FST

Pre-treatment with bicuculline partly prevented the *R*-PHQ- and *S*-PHQ-induced reduction of immobility time in the FST (Figure 4C). Both *R*-PHQ and *S*-PHQ elevated the brain level of GABA, while the GABAergic system antagonist bicuculline attenuated their upregulating effects on brain GABA (Figure 4D).

#### 2.5.3. Involvement of the Noradrenergic System in the Antidepressant-Like Effect of *R*-PHQ and *S*-PHQ in the FST

Pre-treatment with prazosin partly prevented *R*-PHQ- but not *S*-PHQ-induced reduction of immobility time in the FST (Figure 4E). Moreover, *R*-PHQ increased the brain NA content as compared to the control group, while the noradrenergic system antagonist prazosin inhibited such a trend (Figure 4F).

## 3. Discussion

In this study, we firstly reported the antidepressive effects of Rh_2_, *R*-PHQ, and *S*-PHQ using LPS-induced depression-like behavior in FST and TST. All three ginsenosides exhibited clear-cut antidepressive effects characterized by reducing immobile time in FST and TST.

The FST and TST are two widely recognized animal models for the evaluation of antidepressant pharmacological activity [23]. In FST and TST, the animals would become immobile after some struggle because of the stressful environment. Although FST and TST are sensitive to drug administration, sometimes drug-caused enhancement of motor activity may result in a false positive reaction [24]. To rule out this possibility, all mice were subjected to the open-field test. The results showed that Rh_2_, *R*-PHQ, and *S*-PHQ did not change the autonomous activity of mice, and led to the antidepressive-like effects on LPS-challenged mice in FST and TST. The BDNF signaling pathway is a key transducer of antidepressant effects [25]. A low level of BDNF is associated with the pathophysiology of MDD, and BDNF exerts neuroprotective function via its specific high-affinity receptor TrkB [4]. Herein, we found that LPS challenge triggered the decrease of protein levels of brain BDNF and TrkB; however, pre-treatment with Rh_2_, *R*-PHQ, or *S*-PHQ reversed these changes to a relative normal degree at different levels. These observations suggested that Rh_2_, *R*-PHQ, and *S*-PHQ exerted neuroprotective functions via the BDNF signaling pathway in the LPS-induced immunoreactive model.

LPS-induced MDD involves dysfunction of the immune system, which is characterized by abnormal inflammation reactions and an oxidative stress response of the organism [26]. Growing evidence shows that NF-κB regulates hippocampal neurogenesis, which is relevant to mood disorders and the antidepressant activity of drugs [27]. LPS challenge triggered the inflammation response and led to the exposure of the NF-κB p65 subunit, which resulted in the phosphorylation of NF-κB and IκB-α. The released NF-κB bound to corresponding inflammation-associated genes, leading to the excessive generation of IL-6 and TNF-α. The proinflammatory mediators, TNF-α and IL-6, were related to the etiologies of depression. In this study, obviously escalatory amounts of these cytokines were found in the serum of LPS-challenged mice as compared to the control group. However, Rh_2_, *R*-PHQ, or *S*-PHQ treatment clearly reduced the content of TNF-α and IL-6 in a dose-dependent manner, which suggested that the LPS-triggered inflammation response was inhibited. Western blot results were in line with the inhibitory effects of ginsenosides on increased cytokine secretion, whereby Rh_2_, *R*-PHQ or *S*-PHQ treatment inhibited LPS-induced phosphorylation of NF-κB and IκB-α. In brain tissues, LPS exposure caused a marked decrease of Sirt1 and Nrf2, resulting in suppressed levels of antioxidant enzyme SOD [28], which proved the occurrence of an LPS-induced oxidative stress response. However, Rh_2_, *R*-PHQ, or *S*-PHQ treatment effectively reversed these changes. The above-stated phenomena clearly demonstrate that the antidepressant effects of Rh_2_, *R*-PHQ, or *S*-PHQ might be involved in their anti-inflammatory and anti-oxidative properties.

As an earlier report suggested, the antidepressive effects of Rh_2_ were involved in the regulation of DA and NE systems. We used corresponding receptor antagonists in FST to investigate the relationship of *R*-PHQ and *S*-PHQ with DA, GABAergic, and NE systems, and detected the levels of DA, GABA, and NE in mice brain. Finally, we found that the antidepressant effects of *R*-PHQ were mediated by modification of the DA, GABAergic, and NA systems, while the antidepressive effects of *S*-PHQ were mediated by the DA and GABAergic systems only.

The metabolites *R*-PHQ and *S*-PHQ exhibited better antidepressive effects on the immobile time in two behavior tests (FST and TST) than Rh_2_. It was supposed that the existence of the furan ring at C-17 may enhance the antidepressive effects, and the different configurations of C-24 also affected their pharmacological activity, which further proved the opinion that drug metabolites have relatively higher pharmacodynamic effects than prototype drugs. Some other dammarane-type ginseosides such as Re, Rb_1_, and Rg_3_ also have obvious antidepressive effects [11,29,30]. Moreover, it was reported that *R*-PGQ and *S*-PGQ, the cyclization products of Rg_3_, could be synthesized via an oxidizing reaction [31]; however, their antidepressive effects remain unknown and deserve further research.

## 4. Materials and Methods

### 4.1. Preparation of Experimental Drugs

#### 4.1.1. Plant Material

The stems and leaves of *Panax ginseng* were collected from Fusong County of Jilin Province China, in August 2018, and were authenticated by Professor Jinping Liu. A voucher of the specimen (No. SLPG-1808C) was deposited at the National and Local United Engineering Research and Development (R&D) Center of Ginseng Innovative Drugs, China.

#### 4.1.2. Alkaline Hydrolysis of the Total Ginsenosides

The stems and leaves of *Panax ginseng* were hydrolyzed in alkaline propylene glycol solution at 180 °C for 19 h, and then the reaction solution was poured into six volume equivalents of water. The mixture was stirred evenly until the temperature of the suspension lowered successfully to room temperature, and then it was applied to filtration.

#### 4.1.3. Isolation of Rh_2_

The undissolved substance in Section 4.1.2. was subjected to silica gel column chromatography (CC) (200–300 mesh) and was eluted with an acetic ether–methanol (gradient 40:1 to 10:1), whereby Rh_2_-containing fractions were combined with the help of thin-layer chromatography (TLC) detection and were concentrated under vacuum, before being recrystallized in H_2_O–methanol (1:9). The purity of the final product of Rh_2_ was detected via HPLC to be greater than 98.0%.

#### 4.1.4. Synthesis of *R*-PHQ and *S*-PHQ

*R*-PHQ and *S*-PHQ, a pair of Rh_2_-derivative C24-epimers, were semi-synthesized via an oxidization reaction (Figure 5). Additionally, the purity of *R*-PHQ and *S*-PHQ was detected via HPLC to be greater than 98.0%. Briefly, 8.0 g of Rh_2_ was dissolved in 1,4-dioxane, and the solution pH was adjusted to 4.0–5.0. Then, hydrogen peroxide was added to the mixture, which was stirred at 60 °C until the raw material disappeared. Then, the reaction mixture was neutralized with a saturated potassium carbonate solution and was subsequently filtered. The filtrate was concentrated under a vacuum and then loaded onto an ODS column. In total, 3.21 g of *R*-PHQ (yield 40.1%) and 2.76 g of *S*-PHQ (yield 34.5%) were obtained after ODS column chromatography separation (methanol: 60–90%).

#### 4.1.5. Structure Elucidation of Rh_2_, *R*-PHQ, and *S*-PHQ

The structures of *R*-PHQ and *S*-PHQ were identified based on ^1^H NMR, ^13^C NMR, and MS data and by comparison with literature [31].

Structural information of Rh_2_: 3-*O*-β-d-glucopyranosyl-dammar-24-ene-3β, 12β, 20*S*-triol; white powder, C_36_H_62_O_8_, ESI-MS: *m*/*z* 623.5 [M + H]^+^. ^1^H NMR (C_5_D_5_N, 500 MHz) δ: 4.94 (d, *J* = 7.5 Hz, 1H), 4.58 (dd, *J* = 7.5, 2.5 Hz, 1H), 4.39 (m, 1H), 4.22–4.25 (m, 2H), 4.00–4.03 (m, 2H), 3.91 (td, *J* = 10.0, 5.0Hz, 1H), 3.38 (dd, *J* = 11.5, 4.5 Hz, 1H), 1.63 (s, 3H), 1.61 (s, 3H), 1.42 (s, 3H), 1.31 (s, 3H), 0.98 (s, 3H), 0.95 (s, 6H), 0.78 (s, 3H). ^13^C NMR (C_5_D_5_N, 125.8 MHz) δ: 130.8, 126.4, 107.0, 88.8, 78.8, 78.4, 75.8, 73.0, 71.9, 71.0, 63.1, 56.4, 54.8, 51.7, 50.4, 48.6, 40.0, 39.7, 39.2, 37.0, 35.9, 35.2, 32.1, 31.4, 28.2, 27.1, 26.9, 26.8, 25.8, 23.0, 18.5, 17.7, 17.1, 16.8, 16.4, 15.9.

Structural information of *R*-PHQ: (20*S*,24*R*)-3-*O*-β-d-glucopyranosyl-dammar-20, 24-epoxy-3β, 12β, 25-triol; white powder, C_36_H_62_O_9_, electrospray ionization (ESI)-MS: *m*/*z* 639.5 [M + H]^+^. ^1^H NMR (C_5_D_5_N, 600 MHz) δ: 4.95 (d, *J* = 7.8 Hz, 1H), 4.61 (dd, *J* = 11.7, 2.4 Hz, 1H), 4.42 (dd, *J* = 11.7, 5.4 Hz, 1H), 4.21–4.28 (m, 2H), 4.00–4.07 (m, 2H), 3.95 (dd, *J* = 8.3, 6.8 Hz, 1H), 3.72 (td, *J* = 10.4, 4.6 Hz, 1H), 3.38 (dd, *J* = 11.8, 4.4 Hz, 1H), 1.47 (s, 3H), 1.31 (s, 3H), 1.28 (s, 3H), 1.26 (s, 3H), 0.98 (s, 3H), 0.97 (s, 3H), 0.93 (s, 3H), 0.78 (s, 3H). ^13^C NMR (C_5_D_5_N, 150 MHz) δ: 107.4, 89.1, 87.1, 86.0, 79.1, 78.8, 76.2, 72.2, 71.5, 70.7, 63.4, 56.8, 52.6, 51.1, 50.1, 48.8, 40.4, 40.1, 39.6, 37.3, 35.5, 33.2, 32.8, 32.0, 29.2, 28.5, 28.0, 27.6, 27.3, 27.1, 25.8, 18.8, 18.7, 17.1, 16.9, 15.9.

Structural information of *S*-PHQ: (20S,24S)-3-*O*-β-d-glucopyranosyl-dammar-20, 24-epoxy-3β, 12β, 25-triol; white powder, C_36_H_62_O_9_, ESI-MS: *m*/*z* 639.5 [M + H]^+^. ^1^H NMR (C_5_D_5_N, 600 MHz) δ: 4.97 (d, *J* = 7.8 Hz, 1H), 4.50 (d, *J* = 11.4 Hz, 1H), 4.62 (d, *J* = 9.8 Hz, 1H), 4.43 (dd, *J* = 11.7, 5.5 Hz, 1H), 4.21–4.28 (m, 2H), 4.18 (dd, *J* = 10.9, 5.4 Hz, 1H), 4.02–4.08 (m, 2H), 3.78 (td, *J* = 10.1, 4.6 Hz, 1H), 3.40 (dd, *J* = 11.8, 4.3 Hz, 1H), 1.47 (s, 3H), 1.33 (s, 6H), 1.32 (s, 3H), 1.03 (s, 3H), 1.02 (s, 3H), 0.95 (s, 3H), 0.86 (s, 3H). ^13^C NMR (C_5_D_5_N, 150 MHz) *δ*: 107.4, 89.1, 88.8, 87.4, 79.1, 78.8, 76.2, 72.2, 71.1, 70.4, 63.5, 56.8, 52.6, 50.9, 49.9, 49.9, 40.4, 40.1, 39.7, 37.4, 35.5, 33.1, 33.0, 32.6, 29.4, 29.0, 28.5, 27.4, 27.1, 27.0, 26.2, 18.9, 18.5, 17.1, 17.0, 16.0.

### 4.2. Experimental Animal Model and Drug Treatment

Male Institute of Cancer Research (ICR) mice (18–22 g) were obtained from Changsheng Biotechnology Co., Ltd. (Benxi, Liaoning, China). Mice were raised in a standard lab environment (12-h light/dark cycle, 23 ± 1 °C, relative humidity: 50 ± 5%) with free access to food and water, and they were then fasted 12 h prior to the experiment with free access to water only. All experiments were carried out strictly according to the Principle of Laboratory Animal Care and the guidelines prescribed by the Animal Research Committee of Institute of Special Animals and Plants Sciences, Chinese Academy of Agricultural Sciences (Permit No.: ECLA-ISAP-18033).

Experiment 1: After one-week acclimatization, animals were randomly divided into 12 groups (*n* = 10): (1) control, (2) LPS (Sigma-Aldrich, St. Louis, MO, USA), (3) LPS + fluoxetine (FLU, Sigma-Aldrich, St. Louis, MO, USA) (20 mg/kg), (4) LPS + Rh_2_ low dose (L) (7.5 mg/kg), (5) LPS + Rh_2_ medium dose (M) (15 mg/kg), (6) LPS + Rh_2_ high dose (H) (30 mg/kg), (7) LPS + *R*-PHQ-L (7.5 mg/kg), (8) LPS + *R*-PHQ-M (15 mg/kg), (9) LPS + *R*-PHQ-H (30 mg/kg), (10) LPS + *S*-PHQ-L (7.5 mg/kg), (11) LPS + *S*-PHQ-M (15 mg/kg), and (12) LPS + *S*-PHQ-H (30 mg/kg). All mice in drug treatment groups were orally administered with the corresponding dose of drugs that were suspended in 0.05% carboxymethylcellulose sodium (CMC-Na), while the control group and LPS group were orally administered 0.05% CMC-Na once daily for seven consecutive days. On day 7, mice received a single LPS (0.83 mg/kg, intraperitoneally (i.p.)) or 0.9% NaCl aqueous injection 0.5 h after the last drug administration.

Spontaneous locomotor activity test: At day 6, each mouse was placed in the open-field experimental video analysis system placed within a darkened and silent environment, and the number of crossings in the open-field test was recorded over a 5 min period.

Forced swimming test (FST): 24 h after LPS challenge, each mouse was gently placed in an open beaker (height 25 cm, diameter 15 cm) with 20-cm-deep water at 24 ± 2 °C, and forced to swim for 6 min; the total duration of immobility, characterized by stopping struggling and floating motionless on the water during the last 4 min, was recorded by two blinded observers.

Tail suspension test (TST): 26 h after LPS challenge, the tail of the mouse about 1 cm from the end was fixed at the folder, such that it hung from the ground about 50 cm on the bar. Each mouse was suspended for 6 min, and the sum of the immobility time was observed within the last 4 min. After TST, the whole blood of mice was collected, and the serum was separated by centrifugation (3500 rpm, 15 min, 4 °C) and stored at −20 °C; the hippocampus was rapidly collected and stored at −80 °C.

Experiment 2: After one-week acclimatization, animals were randomly divided into 13 groups (*n* = 10): (1) control, (2) FLU (20 mg/kg), (3) *R*-PHQ (30 mg/kg), (4) *S*-PHQ (30 mg/kg), (5) prazosin (Sigma-Aldrich, St. Louis, MO, USA) (1.0 mg/kg), (6) bicuculline (Sigma-Aldrich, St. Louis, MO, USA) (4.0 mg/kg), (7) haloperidol (Sigma-Aldrich, St. Louis, MO, USA) (0.2 mg/kg), (8) prazosin + *R*-PHQ, (9) bicuculline + *R*-PHQ, (10) haloperidol + *R*-PHQ, (11) prazosin + *S*-PHQ, (12) bicuculline + *S*-PHQ, and (13) haloperidol + *S*-PHQ. From day 1 to day 7, prazosin, bicuculline, or haloperidol was administered 30 min prior to *R*-PHQ or *S*-PHQ administration. On day 7, 0.05% CMC-Na, FLU, *R*-PHQ, or *S*-PHQ was administered 30 min prior to FST. The FST was performed as described in Experiment 1. After FST, the brains of mice were rapidly collected and stored at −80 °C.

### 4.3. Enzyme-Linked Immunosorbent Assay

The concentrations of serum TNF-α and IL-6 were measured using ELISA kits (Invitrogen Co., Ltd., Carlsbad, CA, USA) according to the manufacturer’s protocol. The levels of brain monoamine neurotransmitters NA, DA, and GABA were measured using ELISA kits (R&D, Ltd., Minneapolis, MN, USA) according to the manufacturer’s protocol.

### 4.4. Assessment of Biochemical Parameter

The hippocampus tissues were homogenized in 0.9% NaCl and centrifuged (10,000 rpm, 10 min, 4 °C). The supernatant for measuring SOD was applied according to the manufacturer’s protocol (Nanjing Jiancheng Bioengineering Institute, Nanjing, China).

### 4.5. Western Blot

Hippocampus tissues were lysed using radioimmunoprecipitation assay (RIPA) buffer; then, the proteins were electrophoresed on 10% SDS polyacrylamide gels and transferred to a polyvinylidene fluoride (PVDF) membrane. After being blocked for 2 h, the membrane was incubated with primary antibodies against BDNF, TrkB, Sirt1, Nrf2, phosphorylated (*p*)-IκBα, IκB-α, *p*-NF-κB p65, and NF-κB p65 (Cell Signaling Technology, Danvers, MA, USA) overnight at 4 °C. Next, the membranes were incubated for 1 h using secondary antibodies at 37 °C. Finally, the signals were inspected using an enhance chemiluminescence (ECL) substrate. The intensity of the bands was analyzed with Image-Pro Plus 6.0 (Media Cybernetics Inc., Rockville, MD, USA).

### 4.6. Statistical Analysis

Data were analyzed with GraphPad Prism 6.0 software (GraphPad Software Inc., San Diego, CA, USA) and are presented as means ± SD. Statistical significance was calculated with one-way analysis of variance and a *p*-value < 0.05 was considered as significant.

## 5. Conclusions

This study demonstrates that Rh_2_, *R*-PHQ, and *S*-PHQ exhibited antidepressive effects against LPS-induced MDD via a mechanism involving Sirt1/NF-κB and BDNF/TrkB signaling pathways. In summary, pretreatment with Rh_2_, *R*-PHQ, and *S*-PHQ evidently reduced immobile time in the FST and TST, alleviated the LPS-caused inflammation, and recovered the impaired antioxidant system. In addition, the antidepressive effect of *R*-PHQ was involved in DA, GABAergic, and NE systems, while the antidepressive effect of *S*-PHQ was mediated by the DA and GABAergic systems.

## Figures and Tables

**Figure 1 molecules-24-00870-f001:**
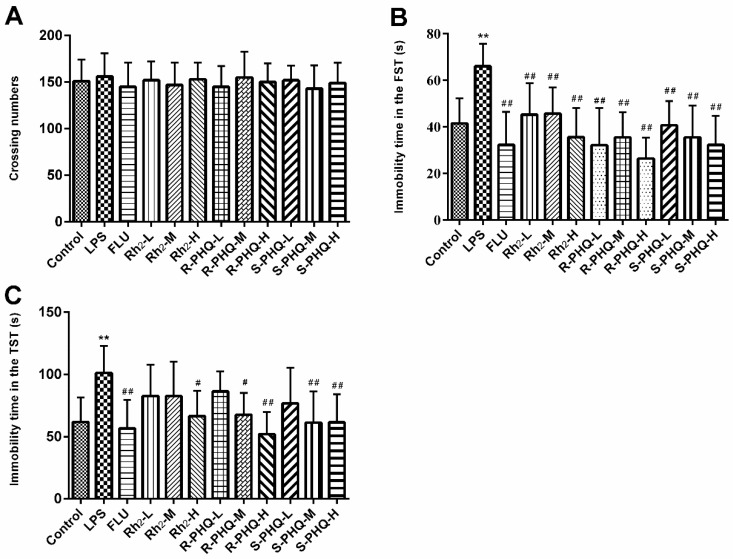
Effects of Rh_2_, (24*R*)-pseudo-ginsenoside HQ (*R*-PHQ), and (24*S*)-pseudo-ginsenoside HQ (*S*-PHQ) on lipopolysaccharide (LPS)-induced depressive-like behavior at a low dose (L, 7.5 mg/kg), medium dose (M, 15 mg/kg), or high dose (H, 30 mg/kg). (**A**) The number of crossings of mice in the open-field test was not affected by LPS exposure, or the administration of Rh_2_, *R*-PHQ, and *S*-PHQ. Immobility time in the (**B**) forced swimming test (FST) and (**C**) tail suspension test (TST) 24 h post LPS challenge, and post administration of Rh_2_, *R*-PHQ, or *S*-PHQ ameliorated LPS-induced depressive-like behavior. Data are expressed as means ± SD (*n* = 10); ** *p <* 0.01 as compared to the control group; ^#^
*p <* 0.05, ^##^
*p <* 0.01 as compared to the LPS-treated group.

**Figure 2 molecules-24-00870-f002:**
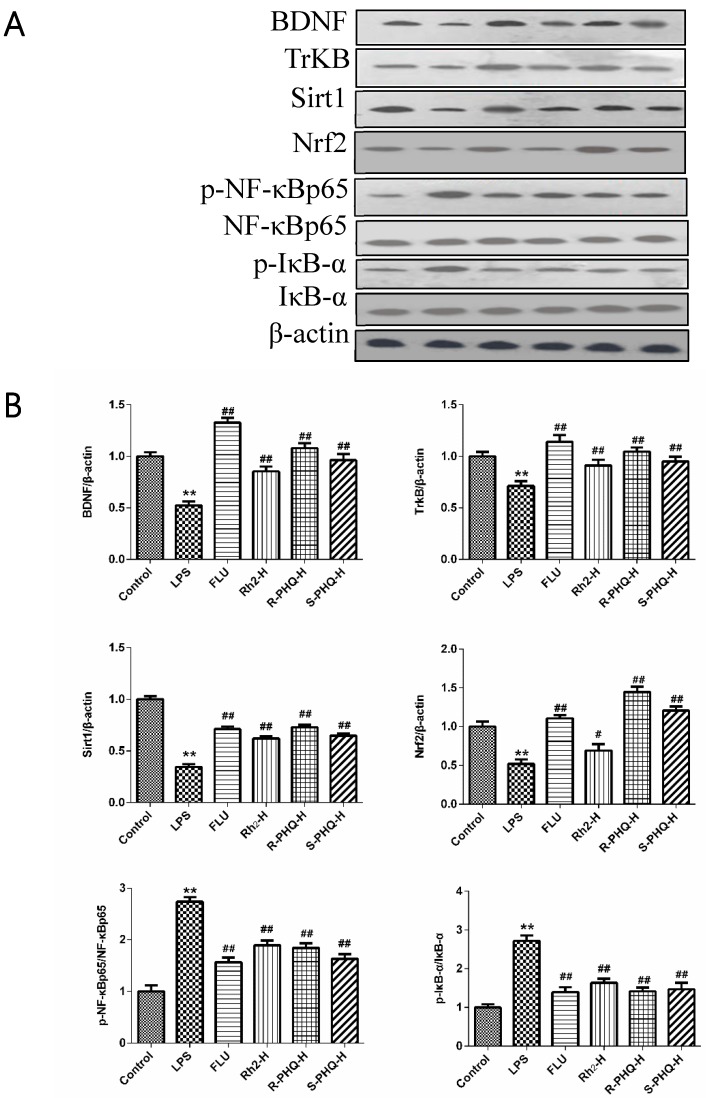
(**A**) Effects of Rh_2_, *S*-PHQ, and *R*-PHQ on the protein expressions of brain-derived neutrophic factor (BDNF), tropomyosin-related kinase B (TrkB), sirtuin type 1 (Sirt1), nuclear-related factor 2 (Nrf2), phosphorylated nuclear factor-κB (*p*-NF-κB) p65, and phosphorylated inhibitor of κB-α (*p*-IκB-α) in mice hippocampus tissues; (**B**) quantitative analyses of BDNF, TrkB, Sirt 1, Nrf2, *p*-NF-κBp65, and *p*-IκB-α in each group. Data are expressed as means ± SD (*n* = 3); ** *p <* 0.01 as compared to the control group; ^#^
*p <* 0.05, ^##^
*p <* 0.01 as compared to the LPS-treated group.

**Figure 3 molecules-24-00870-f003:**
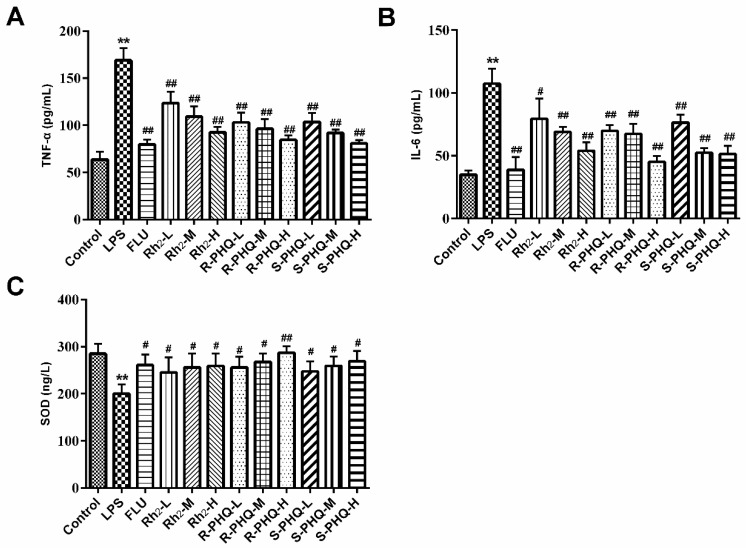
Effects of Rh_2_, *S*-PHQ, and *R*-PHQ on the levels of serum neuroinflammation cytokines (**A**) tumor necrosis factor-alpha (TNF-α) and (**B**) interleukin 6 (IL-6). (**C**) Effects of Rh2, *S*-PHQ, and *R*-PHQ on the level of superoxide dismutase (SOD) in the hippocampus. LPS-triggered inflammation responses and oxidative stress were significantly inhibited by the drug intervention of Rh_2_, *S*-PHQ, or *R*-PHQ. Data are expressed as means ± SD (*n* = 8); ** *p <* 0.01 as compared to the control group; ^#^
*p <* 0.05, ^##^
*p <* 0.01 as compared to the LPS-treated group.

**Figure 4 molecules-24-00870-f004:**
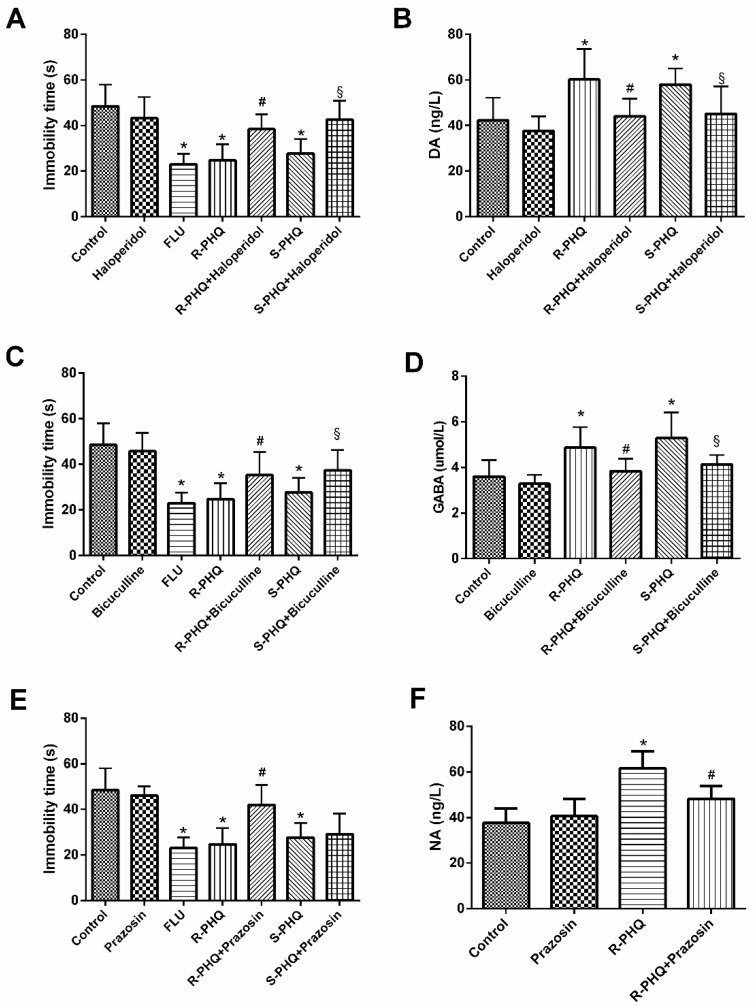
Effects of pretreatment with (**A**) haloperidol (0.2 mg/kg, intraperitoneally (i.p.)), (**C**) bicuculline (4.0 mg/kg), and (**E**) prazosin (1.0 mg/kg) on the antidepressant-like effect induced by *R*-PHQ and *S*-PHQ (30 mg/kg) in the FST; the values are represented as means ± SD (*n* = 10). The effects of *R*-PHQ, *S*-PHQ, haloperidol, bicuculline, prazosin, and their co-treatment on the brain levels of (**B**) dopaminergic (DA), (**D**) γ-aminobutyric acid (GABA), and (**F**) noradrenergic (NA) systems; the values are represented as means ± SD (*n* = 8); * *p <* 0.01 as compared to the control group; ^#^
*p <* 0.01 as compared to the *R*-PHQ-treated group; ^§^
*p <* 0.01 as compared to the *S*-PHQ-treated group.

**Figure 5 molecules-24-00870-f005:**
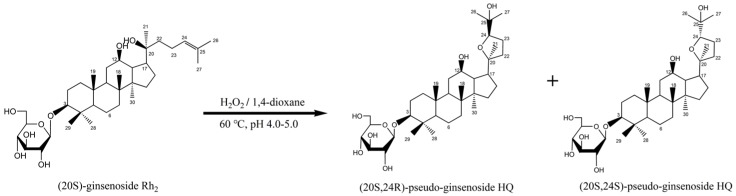
Synthesis process of *R*-PHQ and *S*-PHQ. Rh_2_ was dissolved in 1,4-dioxane at the pH of 4.0–5.0; afterward, hydrogen peroxide was added and the reaction solution was heated at 60 °C to produce the two cyclization products—*R*-PHQ and *S*-PHQ.

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
