# Peer review of "Study on Antidepressant Activity of Pseudo-Ginsenoside HQ on Depression-Like Behavior in Mice"

_molecules, 2019, doi:10.3390/molecules24050870_

Round 1

Reviewer 1 Report

Chen et al. investigated the suppressive effects of ginsenoside Rh2 (Rh2), 24R-pseudoginsenoside HQ (R-PHQ) and 24S-pseudoginsenoside HQ (S-PHQ) against Lipopolysaccharide (LPS)-induced depression-like behavior was evaluated by using the forced swimming test (FST) and tail suspension test (TST) in mice. Overall, this study is interesting, and the data presented in this study are informative. However, several concerns are raising, as listed below. I hope the comments will be helpful to improve the quality of this study.

1. Please state the conclusion of this study at the end of the Abstract chapter.

2. What is DA in the Abstract chapter?

3. Please describe in more detail the relationship between LPS-induced MDD and inflammatory responses as well as oxidative responses in the Introduction chapter. This will help the readers understand the rationale of the experiments in this study.

4. The readers cannot follow what are Rh2-L, Rh2-M, and Rh2-H in the figures. The authors should first describe what are these in the results or in the figure legends.

5. Result 2.1: Use full name of FLU first and then use the abbreviation thereafter.

6. Result 2.2: Please provide the reference(s) for the statement, “TST and FST are two classical behavioral models to evaluate the anti-depressive activity of drugs.”

7. Figure 3: ‘A’ should be given from the Western blot result, and the bar graphs should be labeled from ‘B’.

8. Result 2.4: Please provide the reference(s) for the statement, “Previous studies also reported the involvement of the antioxidant system in the LPS-caused depressant-like behaviors.”

9. 3.5.1. should be 2.5.1.

10. All results should be written in the past tense.

11. Figure legends are too simple, and the authors need to describe the figure legends in more detail.

12. There are many typos (e.g. MMD in line 168 etc...) and grammatical errors. Please go over the entire manuscript and correct all typo and grammatical errors.

Author Response

Document 1. Responses to reviewer 1’s comments

Chen et al. investigated the suppressive effects of ginsenoside Rh2 (Rh2), 24R-pseudoginsenoside HQ (R-PHQ) and 24S-pseudoginsenoside HQ (S-PHQ) against Lipopolysaccharide (LPS)-induced depression-like behavior was evaluated by using the forced swimming test (FST) and tail suspension test (TST) in mice. Overall, this study is interesting, and the data presented in this study are informative. However, several concerns are raising, as listed below. I hope the comments will be helpful to improve the quality of this study.

Comment1: Please state the conclusion of this study at the end of the Abstract chapter.

Response: Thank you for your suggestion. The findings were summarized in this study at the end of the Abstract chapter (line 28-30).

Comment2: What is DA in the Abstract chapter?

Response: Thank you for your careful check. The full name of DA is dopaminergic, and has been added in the Abstract chapter (line 27).

Comment3: Please describe in more detail the relationship between LPS-induced MDD and inflammatory responses as well as oxidative responses in the Introduction chapter. This will help the readers understand the rationale of the experiments in this study.

Response: Thank you, dear reviewer. Your suggestion is very meaningful and far-sighted for the deep study of (LPS)-induced depression-like behavior. The manuscript has described in more detail the relationship between LPS-induced MDD and inflammatory responses as well as oxidative responses in the Introduction chapter (line 52-57 ).

Comment4: The readers cannot follow what are Rh2-L, Rh2-M, and Rh2-H in the figures. The authors should first describe what are these in the results or in the figure legends.

Response: We have rewrite the figure legend of FIGURE 1 as: “Effects of Rh2, R-PHQ, and S-PHQ on LPS-induced depressive-like behavior at low dose (L, 7.5 mg/kg), medium dose (M, 15 mg/kg), or high dose (H, 30 mg/kg)”.

Comment5: Result 2.1: Use full name of FLU first and then use the abbreviation thereafter.

Response: Thank you for your careful check. The full name of FLU is Fluoxetine hydrochloride and has been added in Result 2.1 (line 82).

Comment6: Result 2.2: Please provide the reference(s) for the statement, “TST and FST are two classical behavioral models to evaluate the anti-depressive activity of drugs.

Response: Thank you, dear reviewer. We have provide the correlation reference(s) for the statement, “TST and FST are two classical behavioral models to evaluate the anti-depressive activity of drugs (line 87).

Comment7: Figure 3: ‘A’ should be given from the Western blot result, and the bar graphs should be labeled from ‘B’.

Response: We have do the modification as you suggested. Thanks.

Comment8: Result 2.4: Please provide the reference(s) for the statement, “Previous studies also reported the involvement of the antioxidant system in the LPS-caused depressant-like behaviors.”

Response: Thank you, dear reviewer. We have provide the correlation reference(s) for the statement, “Previous studies also reported the involvement of the antioxidant system in the LPS-caused depressant-like behaviors.”(line 121)

Comment9: 3.5.1. should be 2.5.1.

Response: Thank you for your careful check. We have corrected the error, which 3.5.1 was changed to 2.5.1.

Comment10: All results should be written in the past tense.

Response: Dear reviewer, Thank you for your careful check. All results have been written in the past tense.

Comment11: Figure legends are too simple, and the authors need to describe the figure legends in more detail.

Response: The Figure legends were detailed described. Thanks.

Comment12: There are many typos (e.g. MMD in line 168 etc...) and grammatical errors. Please go over the entire manuscript and correct all typo and grammatical errors.

Response: Thank you for your careful check. We have gone over the entire manuscript and correct all typo and grammatical errors.

Reviewer 2 Report

The aim of the paper was to investigate the effects of ginsenoside Rh2, isolated from stems and leaves of Panax ginseng, and its metabolites (R-PHQ and S-PHQ) on LPS-induced depression in mice. The idea of the work is new, therefore, the work presents a great potential. The experimental part lacks, however, some data important for repetition of the work and verification of its scientific soundness, like the description of the plant material, preparation of alkaline hydrolysate from stems and leaves of Panax ginseng, and detailed characterization of Rh2 isolation from plant extract, and its structure elucidation (identification data), since the identification data for two metabolites of Rh2, i.e. R-PHQ and S-PHQ, has been given. In my opinion, the submission requires minor edition and improvement at some points, main of which are listed below. Materials and Methods: The paragraph 4.1. (Preparation of R-PHQ and S-PHQ) should be replaced by five additional paragraphs, i.e.: 4.1. Plant material 4.2. Preparation of extract (alkaline hydrolysate) 4.3. Isolation of Rh2 4.4. Synthesis of R-PHQ and S-PHQ 4.5. Structure elucidation of Rh2, R-PHQ and S-PHQ

Author Response

Document 2. Responses to reviewer 2’s comments

The aim of the paper was to investigate the effects of ginsenoside Rh2, isolated from stems and leaves of Panax ginseng, and its metabolites (R-PHQ and S-PHQ) on LPS-induced depression in mice. The idea of the work is new, therefore, the work presents a great potential.

Comment: The experimental part lacks, however, some data important for repetition of the work and verification of its scientific soundness, like the description of the plant material, preparation of alkaline hydrolysate from stems and leaves of Panax ginseng, and detailed characterization of Rh2 isolation from plant extract, and its structure elucidation (identification data), since the identification data for two metabolites of Rh2, i.e. R-PHQ and S-PHQ, has been given. In my opinion, the submission requires minor edition and improvement at some points, main of which are listed below. Materials and Methods: The paragraph 4.1. (Preparation of R-PHQ and S-PHQ) should be replaced by five additional paragraphs, i.e.: 4.1. Plant material 4.2. Preparation of extract (alkaline hydrolysate) 4.3. Isolation of Rh2 4.4. Synthesis of R-PHQ and S-PHQ 4.5. Structure elucidation of Rh2, R-PHQ and S-PHQ.

Response: Thank you for your kind suggestion for the scientific soundness of our manuscript. As you suggested, we have reorganized the former paragraph 4.1 and provided the information of the Plant Material and the detail preparation process of ginsenosides as following (the red marked sentences are updated version): 

4.1. Preparation of Experimental Drugs

4.1.1. Plant Material

The stems and leaves of Panax ginseng were collected from Fusong County of Jilin Province China, in August 2018, and authenticated by Professor Jinping Liu. A voucher of the specimen (No. SLPG-1808C) was deposited at the National and Local United Engineering R&D Center of Ginseng Innovative Drugs, China.

4.1.2. Alkaline Hydrolysis of the Total Ginsenosides

    The stems and leaves of Panax ginseng was hydrolyzed in alkaline propylene glycol solution at 180 for 19 h, and then the reaction solution was poured into 6 volumes water. Stirred evenly until the temperature of suspension lowered successfully to room temperature, and then applied to filtration.

4.1.3. Isolation of Rh2

    The undissolved substance in 4.1.2. was subjected to silica gel CC (200-300 mesh) and eluted with acetic ether-methanol (gradient 40:1 to 10:1), Rh2 contained fractions were combined with the help of TLC detection and concentrated under vacuum, then recrystallized in H2O-methanol (1:9). The purity of final product of Rh2 was detected by HPLC to be greater than 98.0%.

4.1.4. Synthesis of R-PHQ and S-PHQ

R-PHQ and S-PHQ, a pair of Rh2 derivative C24-epimers, were semi-synthesized by oxidization reaction (Figure. 5). Additionally, the purity of R-PHQ and S-PHQ was detected by HPLC to be greater than 98.0%. Briefly, 8.0 g Rh2 was dissolved in 1, 4-dioxane, and the solution pH was adjusted to 4.0-5.0. Then hydrogen peroxide was added to the mixture and then was stirred at 60 until the raw material disappeared. Then the reaction mixture was neutralized with a saturated potassium carbonate solution and subsequently filtered. The filtrate was concentrated under a vacuum and then loaded on to an ODS column. 3.21 g R-PHQ (yield 40.1%) and 2.76 g S-PHQ (yield 34.5%) were obtained after ODS column chromatography separation (methanol: 60%-90%).

4.1.5. Structure Elucidation of Rh2, R-PHQ and S-PHQ 

The structures of R-PHQ and S-PHQ were identified based on 1H NMR, 13C NMR, and MS data and by comparison with literature [31].

Structure information of R-PHQ: (20S,24R)-3-O-β-D-glucopyranosyl-dammar-20, 24-epoxy-3β, 12β, 25-triol; white powder, C36H62O9, ESI-MS: m/z 639.5 [M+H]+. 1H NMR (C5D5N, 600 MHz) δ: 4.95 (d, J = 7.8 Hz, 1H), 4.61 (dd, J = 11.7, 2.4 Hz, 1H), 4.42 (dd, J = 11.7, 5.4 Hz, 1H), 4.21-4.28 (m, 2H), 4.00-4.07 (m, 2H), 3.95 (dd, J = 8.3, 6.8 Hz, 1H), 3.72 (td, J = 10.4, 4.6 Hz, 1H), 3.38 (dd, J = 11.8, 4.4 Hz, 1H), 1.47 (s, 3H), 1.31 (s, 3H), 1.28 (s, 3H), 1.26 (s, 3H), 0.98 (s, 3H), 0.97 (s, 3H), 0.93 (s, 3H), 0.78 (s, 3H). 13C NMR (C5D5N, 150 MHz) δ: 107.4, 89.1, 87.1, 86.0, 79.1, 78.8, 76.2, 72.2, 71.5, 70.7, 63.4, 56.8, 52.6, 51.1, 50.1, 48.8, 40.4, 40.1, 39.6, 37.3, 35.5, 33.2, 32.8, 32.0, 29.2, 28.5, 28.0, 27.6, 27.3, 27.1, 25.8, 18.8, 18.7, 17.1, 16.9, 15.9.

Structure information of S-PHQ: (20S,24S)-3-O-β-D-glucopyranosyl-dammar-20, 24-epoxy-3β, 12β, 25-triol; white powder, C36H62O9, ESI-MS: m/z 639.5 [M+H]+. 1H NMR (C5D5N, 600 MHz) δ: 4.97 (d, J = 7.8 Hz, 1H), 4.50 (d, J = 11.4 Hz, 1H), 4.62 (d, J = 9.8 Hz, 1H), 4.43 (dd, J = 11.7, 5.5 Hz, 1H), 4.21-4.28 (m, 2H), 4.18 (dd, J = 10.9, 5.4 Hz, 1H), 4.02-4.08 (m, 2H), 3.78 (td, J = 10.1, 4.6 Hz, 1H), 3.40 (dd, J = 11.8, 4.3 Hz, 1H), 1.47 (s, 3H), 1.33 (s, 6H), 1.32 (s, 3H), 1.03 (s, 3H), 1.02 (s, 3H), 0.95 (s, 3H), 0.86 (s, 3H). 13C NMR (C5D5N, 150 MHz) δ: 107.4, 89.1, 88.8, 87.4, 79.1, 78.8, 76.2, 72.2, 71.1, 70.4, 63.5, 56.8, 52.6, 50.9, 49.9, 49.9, 40.4, 40.1, 39.7, 37.4, 35.5, 33.1, 33.0, 32.6, 29.4, 29.0, 28.5, 27.4, 27.1, 27.0, 26.2, 18.9, 18.5, 17.1, 17.0, 16.0.

Structure information of Rh2: 3-O-β-D-glucopyranosyl-dammar-24–ene-3β, 12β, 20S-triol; white powder, C36H62O8, ESI-MS: m/z 623.5 [M+H]+. 1H NMR (C5D5N, 500 MHz) δ: 4.94 (d, J = 7.5 Hz, 1H), 4.58 (dd, J = 7.5, 2.5 Hz, 1H), 4.39 (m, 1H), 4.22-4.25 (m, 2H), 4.00-4.03 (m, 2H), 3.91 (td, J = 10.0, 5.0Hz, 1H), 3.38 (dd, J = 11.5, 4.5 Hz, 1H), 1.63 (s, 3H), 1.61 (s, 3H), 1.42 (s, 3H), 1.31 (s, 3H), 0.98 (s, 3H), 0.95 (s, 6H), 0.78 (s, 3H). 13C NMR (C5D5N, 125.8 MHz) δ: 130.8, 126.4, 107.0, 88.8, 78.8, 78.4, 75.8, 73.0, 71.9, 71.0, 63.1, 56.4, 54.8, 51.7, 50.4, 48.6, 40.0, 39.7, 39.2, 37.0, 35.9, 35.2, 32.1, 31.4, 28.2, 27.1, 26.9, 26.8, 25.8, 23.0, 18.5, 17.7, 17.1, 16.8, 16.4, 15.9.